# Case management programs for people with complex needs: Towards better engagement of community pharmacies and community-based organisations

Maud-Christine Chouinard[1]☯*, Mathieu Bisson[2]☯, Alya Danish[2‡], Marlène Karam[1‡], Jérémie Beaudin[2‡], Nevena Grgurevic[2‡], Véronique Sabourin[3‡], Catherine Hudon[2]☯

**1** Faculty of Nursing, University of Montreal, Montreal, Quebec, Canada, **2** Department of Family Medicine and Emergency Medicine, University of Sherbrooke, Sherbrooke, Quebec, Canada, **3** Centre Intégré Universitaire de Santé et de Services Sociaux (CIUSSS) of the Saguenay-Lac-Saint-Jean, Chicoutimi, Quebec, Canada

☯ These authors contributed equally to this work.
‡ These authors also contributed equally to this work.
* Maud.Christine.Chouinard@umontreal.ca

**Data Availability Statement:** Due to the Canadian regulatory framework (https://ethics.gc.ca/eng/policy-politique_tcps2-eptc2_2018.html), we

## Abstract

### Introduction

The objectives of this study were 1) to describe how case management programs engaged community pharmacies and community-based organisations in a perspective of integrated care for people with complex needs, and 2) to identify enablers, barriers and potential strategies for this engagement.

### Methods

Using a descriptive qualitative design, individual interviews and focus groups with patients, healthcare providers and managers were analysed according to a mixed thematic analysis based on a deductive (Rainbow Model of Integrated Care) and an inductive approach.

### Results and discussion

Participants highlighted the individualized service plan as a significant tool to foster a shared person-focused vision of care, information exchanges and concerted efforts. Openness to collaboration was also considered as an enabler for community stakeholders' engagement. The lack of recognition of community-based organisations by certain providers and the time required to participate in individualized service plans were outlined as barriers to professional integration. Limited opportunities for community stakeholders to be involved in decision-making within case management programs were reported as another constraint to their engagement. Cultural differences between organisations regarding the focus of the intervention (psychosocial vs healthcare needs) and differences in bureaucratic structures and funding mechanisms may negatively affect community stakeholders' engagement. Formal consultation mechanisms and improvement of communication channels between

cannot provide the entirety of the dataset of this study. The consent form based on this framework and signed by the participants contains a section in which they have agreed that their data may be reused, but only as part of a sub-study that must be reviewed and approved by the responsible ethics committee. Researchers who wish to access the data can request it at the following contact information: Comité d'éthique de la recherche / Research Ethic Board Centre intégré universitaire de santé et de services sociaux du Saguenay–Lac-Saint-Jean E-mail: guichetunique.slsj@ssss.gouv. qc.ca Project #2014-015.

**Funding:** MCC and CH received a funding from the Canadian Institutes of Health Research (CIHR), grant number 318771. URL of the funder website: https://cihr-irsc.gc.ca/e/193.html The funder had no role in study design, data collection and analysis, decision to publish, or preparation of the manuscript.

**Competing interests:** The authors have declared that no competing interests exist.

healthcare providers and community stakeholders were suggested as ways to overcome these barriers.

## Conclusion

Efforts to improve care integration in case management programs should be directed toward the recognition of community stakeholders as co-producers of care and co-builders of social policies across the entire care continuum for people with complex needs.

## Introduction

People with complex care needs are characterized by multiple chronic diseases, mental health comorbidities and/or social vulnerabilities [1]. These individuals are at greater risk for adverse health outcomes, reduced quality of life and increased mortality [2, 3]. They constitute a small heterogeneous group (10% of all users) that generates disproportional costs (70%) for the health system in Canada as in many industrialized countries [4, 5]. Their high use of emergency department services and hospitalizations [5–7] is generally due to fragmented and episodic care between healthcare services [8, 9]. As a result, providing appropriate services that meet the needs of this population is required and can be achieved through integrated care. Integrated care is "the search to connect the healthcare system (acute, primary medical, and skilled) with other human service systems (e.g. long-term care, education, and vocational and housing services) in order to improve outcomes (clinical, satisfaction, and efficiency)" [10]. Case management programs (CMPs) are increasingly used to improve the integration of services [11, 12]. Defined as 'collaborative, client-driven processes for the provision of quality health and support services through the effective and efficient use of resources' [13], their benefits for patients include the improvement of self-management skills, adherence, satisfaction, health status and quality of life. CMPs also benefit the healthcare system by improving the quality of care and reducing healthcare use and cost [14–18].

In Quebec (Canada), community-based organisations (CBOs) and community pharmacies are primary care services linked by formal and informal arrangements to healthcare organisations [19].

CBOs are non-profit organisations that work for social development in their communities [20]. and may include volunteer associations, cooperatives and social economy enterprises whose funding comes from various sources (federal and provincial governments, foundations, donations, social economy, etc.) [21]. Their missions focus on social development, advocacy, housing, and recreation. The populations that they target may include youth, families, indigenous groups, LGBTQ+, people with poor mental health or disabilities, refugees, homeless individuals and immigrants. Their intervention approaches are diverse and include health promotion, informal intervention, outreach work, harm reduction, empowerment, group therapy, and person-focused approaches. As local and collective initiatives, the majority of CBOs focus on community needs, with governance that is based on autonomous and democratic principles, usually involving a board of directors mandated by an assembly of representatives who supervises employee activities and the organisation's strategic orientation [22].

Community pharmacies are private organisations committed to maintaining the overall health of their patients through a variety of interventions: medical information review and treatment follow-up, preparation of medication, adjustment and initiation of treatment, and daily consultations with people who have questions [23]. Community pharmacists are

indispensable partners for patients with polypharmacy [24, 25]. They educate and advise people on the use of over-the-counter or prescribed medication and natural health products; support people seeking solutions to minor health problems; contribute to patients' education regarding healthy lifestyles; and if necessary, refer patients to other health and social services [23].

Research has demonstrated the need for and benefits of engaging CBOs and community pharmacies in hospitals or primary care programs, such as CMPs, for people with complex care needs [26–28]. These community stakeholders, who are deeply rooted in their communities, can facilitate a close connection with people who live in the surrounding area [29]. As such, they can contribute significantly to identifying people with complex care needs [27] and to supporting them as they navigate the health system [30]. Furthermore, most CBOs offer person-focused interventions that target behavioural issues or functional difficulties (e.g. harm reduction, education on healthy lifestyle), which may be an effective component of CMPs, especially if linked to healthcare delivery [30]. Fig 1 illustrates the relationships between the stakeholders engaged in these programs for patients with complex needs.

Despite this evidence, the collaboration between providers from the healthcare system and community stakeholders remains poor due to healthcare professionals' lack of knowledge about CBOs and community pharmacies [31], resulting in low CMP referral rates, complexity of managing multi-organisational initiatives, variable adherence to the programs [26] and poor care transition leadership [30]. Disparities in financial resources, organisational expertise and knowledge, lack of proximity between organisations, differences in the vision of collaboration, and complexity of managing communication and information flow across organisations are other issues that can challenge alliances [32].

It is now recognized that CMPs can help bridge the gap between institutional and community care, and that inter-organisational collaboration, as proposed by these programs, could help "service organisations to shift from traditional 'silo' models of service delivery to increased community-based collaboration and service coordination" [33]. Yet, evidence regarding integrated care in the context of CMPs from stakeholders' perspective remains limited [28, 33–35]. The objectives of this study are: 1) to describe how CMPs engaged community pharmacies and CBOs in a perspective of integrated care for people with complex needs, and 2) to identify engagement enablers, barriers, and potential strategies to overcome these barriers.

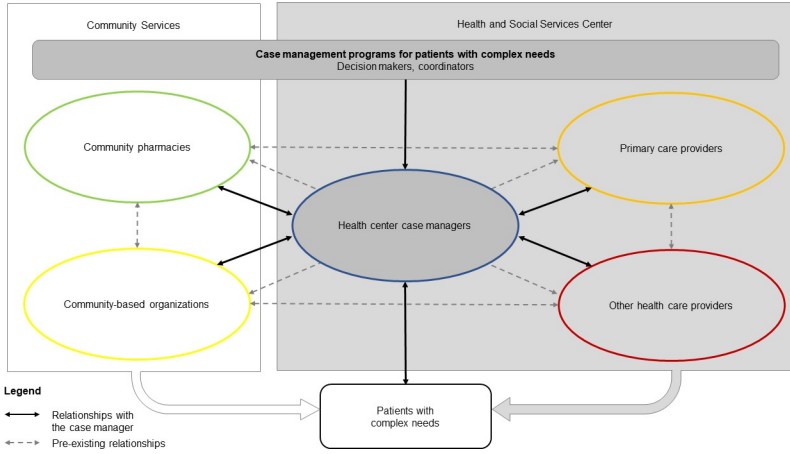

**Fig 1. Relationships between stakeholders engaged in CMPs.**

## Conceptual framework

The Taxonomy for Integrated Care [36] based on the theoretical foundations of the Rainbow Model of Integrated Care [37] was used in this study. The Rainbow Model of Integrated Care plays interconnected roles at the macro (system integration), meso (organisational and professional integration), and micro (clinical integration) levels, as well as between these levels (functional and normative integration). It was developed from electronic database searches, hand searches of reference lists (snowball method) and by contacting researchers in the field [37]. Thereafter, a literature review and thematic analysis procedure were conducted to refine the model into the taxonomy of fifty-nine key features that helps to profile integrated care initiatives [36]. By developing an international consensus-based taxonomy based on Delphi studies [38] and including every level and stakeholders' perspectives of integrated care, Valentijn et al.'s research has become a reference in the field of integrated care.

At the macro level, system integration refers to the alignment of rules and policies within a system to ensure the provision of continuous, comprehensive, and coordinated services across the entire care continuum [37]. At the meso level, organisational and professional integrations refer to the extent to which organisations and healthcare providers respectively coordinate services across organisations and disciplines. These types of integration processes are especially relevant "in socially disadvantaged populations, such as those with large variations in wealth, education, culture and access to healthcare" [39]. At the micro level, clinical integration is related to how care services are coordinated, share a single process for person-focused coordination of care across time, places and disciplines, and reflect a bio-psychosocial perspective of health. This person-focused coordination taking into account the broader health context is particularly relevant for people with complex healthcare needs that span a large number of service areas [37]. Clinical integration also encompasses the important aspect of the patient as a co-creator in the care process and shared responsibility between the provider and the person [40].

Functional and normative integration are cross-cutting types of integrated care processes linking macro, meso, and micro levels [37]. Functional integration links financial, management and information systems around the primary process of service delivery across clinical, professional, organisational and system integration. Normative integration implies the development and maintenance of a common frame of reference (i.e. shared mission, vision, values and culture) between organisations, professional groups and individuals.

## Material and methods

### Design

A descriptive qualitative design [41] was used. This approach helps to provide a full description of the individuals' experience, perceptions, and knowledge of the CMPs, in plain language while remaining close to the data and minimizing researcher influence on data interpretation [42, 43]. It helped to obtain a better understanding of the stakeholders' engagement (strategies, barriers and facilitators) in a perspective of integrated care.

### Settings

This study was developed as part of the developmental evaluation of a CMP in the Integrated University Health and Social Services Centre (hereafter called 'hospital network") located in the Saguenay-Lac-Saint-Jean region of the province of Québec (Canada) [12]. This hospital network is composed of six health and social services centres (hereafter called 'hospital'), each including a hospital, community and long-term care centres, a child and youth protection centre, and a rehabilitation centre to ensure access, continuity, coordination and the quality of

services intended for the population of their local territories [44]. Patients eligible for the CMP had complex needs and had made six or more visits to the emergency department, or had three or more hospitalizations in the previous year. The study was conducted in partnership with hospital network decision makers and a variety of stakeholders [12].

Between 2008 and 2015, the CMP for patients with complex care needs was deployed in all of the hospitals within the hospital network. The program comprised four main components: 1) evaluation of patient needs and goals; 2) development of a patient-centred individualized service plan [45]; 3) care coordination among all partners; and 4) education and self-management support for patients and families [12]. The individualized service plan was planned by the case manager after obtaining the patient's consent. It involved all stakeholders in a meeting, including the patient, primary care providers, secondary and tertiary care providers, community pharmacists, CBO representatives (typically social workers) and the case manager, to detail the patient's needs (including an orientation regarding the action plan for each need) and person-focused shared objectives as well as the services allocated in response to these needs and objectives [45]. The case manager was mandated to validate, share and ensure the follow-up of the individualized service plan with the concerned stakeholders. The aim of the case manager's intervention was to improve quality of life and self-management for patients, and for organisations, it aimed to improve care integration and support for healthcare teams to reduce inappropriate use of services and costs [12].

## Data collection and characteristics of participants

Key informants involved in the CMPs were recruited through purposeful sampling [46] in the six hospitals between December 2014 and May 2018. Patients recruited for the study met the program's eligibility criteria, i.e. frequent users of hospital services who had six or more visits to the emergency department, or three or more hospitalizations in the previous year. Patients were approached by their case manager to participate in the study. Those who verbally consented to participate were referred by their case manager to the research team. The research team members then contacted the patients by phone to make an appointment for an individual interview. The researchers' knowledge of the hospital network's organization helped them identify managers, clinicians and community pharmacists. CBO representatives were identified with the help of case managers [46]. Research assistants explained the research project to them as part of the first contact by phone or email. An appointment was then made for the individual interview or focus group.

Individual interviews and focus groups were both used as qualitative data collection methods to promote participation and facilitate exchanges. Individual interviews aim to thoroughly explore each participant's views, experiences, beliefs, and knowledge, while focus groups use group dynamics to highlight the variation of viewpoints held in the targeted population [47]. Focus groups were used as an alternative method to individual interviews, gathering selected types of actors to facilitate their participation, before or after one of their scheduled meetings. One-hour individual interviews (n = 58 participants) were conducted with people with complex care needs (n = 25), managers, case managers and coordinators (n = 13), family physicians (n = 16), and community pharmacists (n = 4). Focus groups (n = 13, including 71 participants) lasting between 45 and 90 minutes were conducted with managers and case managers (n = 4, including 22 participants), family physicians (n = 2, including 16 participants), nurses (n = 1, including three participants), community pharmacists (n = 2, including five participants) and CBO representatives (n = 4, including 25 participants). Table 1 presents the characteristics of the participants. The focus groups included two to eight participants. A total of seven to nine participants provides a balance between the number of interactions by participants and the

variation of experiences and opinions, while more specialized topics work best with groups of five or six participants [47]. The small size of certain focus groups is due to the unavailability of some participants from the same category to gather at the same time and in the same place. Even in the smaller groups, interactions between the participants produced deeper discussions, thereby improving understanding [47]. Individual interviews and focus groups were conducted face-to-face by four master's level research assistants experienced in qualitative research (two with a background in social work, and two in anthropology). One research assistant facilitated the focus group while another took notes. The semi-structured topic guide used by the research assistants was informed by the literature review (including integration dimensions) and discussions across the research team to achieve the objectives of the study. Questions were adapted to the various categories of participants and validated by the research team members, including a patient partner. The Interview Guide is reported in the S1 File. Individual interviews and focus groups were recorded and transcribed verbatim. Excerpts were anonymized.

**Table 1. Characteristics of the participants (n = 129).**

| Participants | Patients | | Health professionals | Managers |
|---|---|---|---|---|
| **Type of interview: n** | | | | |
| Individual interviews | 25 | | 20 | 13 |
| Focus groups | 0 | | 9 | 4 |
| **Total of participants: n** | 25 | | 69 | 35 |
| **Variables** | | | | |
| Gender: n (%) | | Gender: n (%) | | |
| Female | 12 (48%) | Female | 47 (68%) | 26 (74%) |
| Male | 13 (52%) | Male | 22 (32%) | 7 (20%) |
| Age (years): n (%) | | Years of experience: ($\bar{x}$) | 12 | 7 |
| 18–40 | 3 (13%) | Profession: n (%) | | |
| 41–64 | 10 (33%) | Family physicians | 32 (46%) | |
| 65+ | 12 (50%) | Primary care nurses | 3 (<4%) | |
| Educational level: n (%) | | Pharmacists | 9 (13%) | |
| None | 1 (4%) | Community representatives | 25 (36%) | |
| Primary | 7 (29%) | | | |
| Secondary | 15 (58%) | | | |
| College | 1 (4%) | | | |
| University | 0 (0%) | | | |
| Occupation: n (%) | | | | |
| Full-time/part-time work | 3 (13%) | | | |
| Full-time school | 1 (4%) | | | |
| Unable to work due to health condition | 9 (33%) | | | |
| Retired | 10 (42%) | | | |
| Married | 11 (46%) | | | |
| Single | 8 (33%) | | | |
| Divorced/separated | 4 (13%) | | | |
| Widowed | 1 (4%) | | | |
| Income (CAN$): n (%) | | | | |
| $0–$20,000 | 15 (58%) | | | |
| $20,000–$40,000 | 5 (21%) | | | |
| $40,000-$60,000 | 1 (4%) | | | |
| $60,000–$100,000 | 1 (4%) | | | |

Credibility (accurate description of the phenomenon) was ensured by asking open-ended questions, by allowing participants some latitude in what they wished to reveal and by the triangulation of informants. Data saturation was not targeted for each of the participant categories, but the diversity of the actors involved (triangulation) allowed for a comprehensive representation of the phenomenon and enhanced trustworthiness [41, 48].

## Analysis

Experiences and opinions collected from the participants were analysed according to a mixed thematic analysis [49]. Consistent with the descriptive design, this approach helps to identify "codes" or labels that assign symbolic meaning to the raw descriptive information compiled during the study [49]. Four research team members took part in the analysis process according to three iterative stages allowing data-driven coding and categorization to identify emergent themes and trends: data condensation, data organisation and their interpretation [49]. First, data were categorized in themes identified according to Valentijn et al.'s taxonomy and conceptual framework [36, 37] (deductive) and other relevant information allowing us to achieve the research objectives (inductive). This step of data condensation was processed using NVivo software (Version 11). Second, tables were created to organise and synthesize the data, grouping them into a smaller number of themes (data organisation). Third, patterns were identified, described, and explained (interpretation). Three members of the research team validated each step of the thematic analysis process according to the investigator triangulation method [50].

This study received approval from the Ethics Review Boards of the Saguenay-Lac-Saint-Jean Integrated University Health and Social Services Centres. All informed consent was given in writing.

## Results

Table 2 summarizes the results presented in the following section.

### Clinical integration

Participants recognized the usefulness of the individualized service plan as a powerful tool to ensure a global understanding of the people's situation, focusing on their priorities and enabling the complementarity of health care and psychosocial resources.

> "Sometimes the individualized service plan is where you can really get to know the person a little more as a whole." (Focus group with CBO representatives)

> "Involving them [people with complex needs] as well as making them responsible for the overuse of services; whether by having them attend the individualized service plan meeting, or other such individualized references, I believe is empowering for these people." (Focus group with CBO representatives)

> "We developed an individualized service plan so that all the stakeholders on both the social and physical sides understand the consequences of my health problems and treatment. . . I can explain my background. I know my situation very well." (Individual interview with a patient)

The difficulty for CBOs to help people with physical pain was also mentioned and calls for collaboration between healthcare services, illustrating the complementarity of healthcare and community resources.

**Table 2. Strategies, enablers, and barriers for community stakeholders' engagement in CMPs according to the Rainbow Model of Integrated Care.**

| Integration dimensions | Engagement strategies | Engagement enablers | Engagement barriers |
|---|---|---|---|
| **Clinical**<br>Coordination of person-focused care in a single process across time, place and discipline | Use of the individualized service plan<br>Care coordination by the case manager<br>Person-focused intervention | Patients' involvement<br>Global understanding of the patient<br>Mutual understanding of roles<br>Complementarity of health care and community resources | - |
| **Professional**<br>Inter-professional partnerships | Use of the individualized service plan<br>Inter-professional collaboration | Shared vision, collaboration, and consensus among providers<br>Interdependence between hospital and community stakeholders<br>Less services duplication<br>Less contradictions in care planning | Lack of recognition of CBOs by certain hospital providers<br>Time required to participate in an individualized service plan |
| **Organizational**<br>Inter-organizational partnerships | Formal consultation mechanisms between hospital and CBOs<br>Inter-organizational collaboration<br>Decision makers and managers support | Knowledge of each other organizations involved in the program<br>Concerted efforts | Lack of opportunities for community stakeholders to be involved in decision-making processes within CMPs |
| **Systemic**<br>Policy arrangements | - | - | - |
| **Functional**<br>Support mechanisms and communication tools | Financial, managerial, and informational support<br>Formal communication channels between the hospital and community stakeholders | Access to the patient's information<br>Staff stability<br>Previous collaboration established between the case manager and community stakeholders | Different opening hours from one organization to another |
| **Normative**<br>Cultural frame of reference mutually respected by all | Use of the individualized service plan | Individual openness to collaboration<br>Common purpose towards frequent users of health services | Cultural differences in focus on physical vs psychosocial health<br>Differences in bureaucratic structures and funding mechanisms |

The important role of case managers in care coordination across services at the clinical level has also been raised by a participant.

"It definitely takes a conductor for this global coordination. The case manager is like the orchestra's conductor." (Individual interview with a family physician)

Community stakeholders and hospital providers both recognized and adopted the person-focused approach, which improves clinical integration and may enable collaboration and engagement between stakeholders.

"The minute you hear the words 'vulnerable clientele' [. . .] It means . . . these people have special needs, and someone has to take care of them, no matter who. It may be a team, an individual, several people, a community, a society, a family, whatever [. . .] together with the client, with his or her experience, we will take him or her elsewhere." (Individual interview with a hospital manager)

"You don't have to work for the system, to unclog the system, you have to work for the person. If you focus on that, maybe the solutions will be easier than passing the buck." (Focus group with CBO representatives)

## Professional integration

Two main processes related to professional integration were described by participants: having a shared vision between providers focusing on the content of care and the development of an

interdependence between hospital providers and community stakeholders. Again, the individualized service plan was outlined as an important tool to foster a shared vision, enabling consensus among stakeholders and reducing duplication of services. Community stakeholders recognized the added value of the collaboration. Providers from the hospital and community stakeholders also recognized that individualized service plans support the development of collaboration with CBOs.

> "Everyone is on the same page, everyone has a defined role, rather than sometimes duplicating services or contradicting each other. People cannot always come together, which is what individualized service plans allow them to do." (Focus group with CBO representatives)

> "We are all here to discuss the same patient. It's amazing how together we make a much greater difference than each of us on their own." (Individual interview with a community pharmacist)

However, a condescending attitude toward and lack of recognition of CBOs by certain hospital providers and the time required to participate in an individualized service plan were outlined as barriers to professional integration.

> "In terms of personality, there are some who will come to us and impose themselves as experts. 'Look here, I've been doing this for 25 years. . .' But not everyone is like that. There are others who arrive a little awkwardly, they are great to deal with. So that's it, there is also a lot of whoever you have in front of you." (Focus group with CBO representatives)

> "There is a lack of knowledge about the existence of community services, but once you know about them, you have to recognize the professional expertise within the community network [. . .]." (Focus group with CBO representatives)

> "I mean, it's a barrier that we have to take the time, in community pharmacies, to participate in an individualized service plan. It's a major financial hurdle [. . .]." (Individual interview with a community pharmacist)

## Organisational integration

Organisational integration between healthcare services and community stakeholders in CMPs happened mainly through knowledge of each other and through concerted efforts between these organisations.

> "Of course, it requires a concerted effort, but the providers also need to know about the organisations' services, departments, and missions, whether through us or others. For example, for a patient who never comes to his appointments, because he has atypical hours, he sleeps during the day, there is street work, there are outreach services and community organisations that work at atypical hours, which could help us to remedy the situation as well as taking part in the individualized service plan." (Focus group with CBO representatives)

> "All these organisations [CBOs] are often useful for respite. And, often, when people live in isolation, if they don't know what to do, they come to the emergency department or their level of distress rises quickly. I believe that these organisations do have a complementary role." (Focus group with hospital managers)

"I just wanted to add that there are a lot of community organisations in our mental health individualized service plans [. . .] There is a great collaboration. [. . .]

I've been the coordinator since 2008, and it's amazing how much better our relationship with the community network gets every year." (Focus group with a coordinator and a case manager)

Lack of opportunities for CBOs and community pharmacies to be involved in decision-making processes within CMPs for people with complex care needs was reported as a significant barrier to their engagement. As suggested by some participants, CBOs and community pharmacies should be more involved in decision-making about these programs and especially about the way patients can be identified and supported. Formal consultation mechanisms between the hospital and CBOs were suggested to overcome this barrier.

"[. . .] could a complex case committee not be set up, with the [health] network and with community organisations, so that we can work in collaboration rather than just one way." (Focus group with CBO representatives)

### System integration

No direct processes concerning system integration were identified from the participants' narratives.

### Functional integration

Information management and resource management were the only two functional integra-tion- processes described by the participants. Knowledge of the individualized service plan by CBO representatives and pharmacists provides an overview of the patient's situation, so they can refer the patient or intervene more efficiently. For their part, patients do not have to repeat their stories to every care team member. Some other communication channels between the hospital and community stakeholders were recognized as promoting better access to the patient's information and to common knowledge that facilitates clinical, professional and organisational integration.

However, most of these communication channels relied on previous collaboration between the case manager and stakeholders involved. These narratives illustrate how both information management and resource management can influence functional integration and may demo-bilize stakeholders.

"When we know people and we have a good relationship, we have the right information. When these people retire, change jobs, or leave the organisation, we lose it [. . .]. It is chal-lenging because there is no established communication structure. There should be clear channels of communication and staff management that improve staff stability, but there is nothing, it's case-by-case. [. . .] The turnover rate means it changes all the time." (Focus group with CBO representatives)

### Normative integration

Cultural differences between hospital and community stakeholders regarding the focus of the intervention (psychosocial vs healthcare needs) and differences in bureaucratic structures and funding mechanisms may affect community stakeholders' engagement.

"I would tell you that, with the hospital, of course, we have to work together anyway, the partnership is still going well, but we have to work on it. Because, in fact, it's two different cultures, the way of doing things is different too. Of course, there is dissatisfaction in the way of doing things." (Focus group with CBO representatives)

"There is a reality with regard to CBOs, which is that they are autonomous, they can do what they want, and then the funding, which is related to this among other things, means that, theoretically, we are not required to have relationships. Therefore, it depends more on the goodwill of the people who work there." (Focus group with CBOs representatives)

Despite these differences, the organization of individualized service plans can help stakeholders focus on a common purpose for frequent users of health services.

"What we often realize in the individualized service plans is that we worked in different ways, in different directions, and the person was quite happy with that. Now, when we all go the same way, it's much simpler. The person is well supported, and we know where we are going. It works, it works." (Focus group with CBO representatives)

## Discussion

There is growing recognition that integrating care can improve patients' outcomes, especially among those with complex health and social needs [25]. By ensuring communication and collaboration between professionals of various organisations, and the participation of every stakeholder, case managers are "searching for connections between the healthcare system and other human service systems to improve outcomes", which correspond precisely to the definition of integrated care as stated by Leutz [10]. Previous studies showed that promoting interorganizational collaboration faces a greater challenge than promoting interprofessional collaboration due to differences between corporate cultures, geographical distance, the multitude of processes, and formal paths of communication [51]. The results of this study confirm the gap between community stakeholders and CMPs due to these challenges and offer new insights into this engagement.

CBOs and community pharmacies wish to be engaged in CMPs. Their proximity to the population (physical presence in the living environment), their adaptability and plurality of service delivery, their knowledge of the daily users' situation and individual needs and goals, and their complementary knowledge, whether about pharmaceutical or psychosocial aspects can contribute significantly to improving the programs [52–55]. CBO and pharmacy stakeholders are also well positioned to help identifying people with complex care needs.

For community stakeholders, the individualized service plan remain the main ingredient of the CMP. The use of a multidisciplinary/interorganisational care plan is already recognized as an effective approach to aligning the goals of the different healthcare services and as an effective strategy to ensure positive program outcomes for people with complex health and social needs [56–58]. Community stakeholders believe that they can and should contribute to the individualized service plan. According to them, this contribution could improve global patient engagement, better access to patient information and interprofessional collaboration. However, cultural differences, as well as challenges in communication channels were raised as significant barriers to this contribution, as collaboration still often relies on a history of collaboration between involved parties [26, 37, 55, 59].

In response to these challenges, many participants outlined, as suggested by other authors [60], the importance of formalizing partnerships and communication channels. These

improvements should span "over the full continuum of services as opposed to separate providers and sectors" [61], building on but going beyond previous collaborations. As observed by Fleury et al. (2014), a vulnerable population with complex needs evolving in a decentralized network needs more formal partnerships to improve the integration of services [62]. As one research participant mentioned, "Collaboration must become the norm". Hospital decisions should consider the inclusion of community stakeholders on CMP governance committees. To do this, community-based and person-focused paradigms of care must be strengthened [63] and community stakeholders must be considered as co-producers of care [64] and, ultimately, as co-builders of social policies [21] for people with complex needs. Decision makers must consider adequate funding [60] dedicated to community stakeholders participation in individualized service plans [65, 66] and their engagement in CMPs. In the same way, the programs need to be supported and pursued in a perspective of ongoing improvement.

## Limitations of the study

The community stakeholders who participated in the interviews and focus groups did not all have direct exposure to the CMPs. Their variable experiences within the programs may have influenced the results. However, all community stakeholders were referred to the research team by a case manager, worked with the targeted clientele, had connections with the health and social services network and had a minimum of knowledge about the programs. Their contribution was still relevant to the study.

As mentioned, the Interview Guide was not formatted based on the Rainbow Model of Integrated Care. Although it is a robust framework, elaborated to provide all stakeholders' perspectives at all levels (macro, meso and micro) and internationally recognized in the field of integrated care, it remains difficult to use in the way it is formulated. For example, no system integration processes were identified from the participants' narratives, but this type of integration can be difficult to differentiate from organisational integration and may be less relevant to clinical stakeholders [36, 38].

Finally, the limited description of the settings where CMPs were implemented makes the transferability of the results difficult. However, the heterogeneity of the contexts (i.e. populations served by CMPs, their urban and rural environments, their size, the types of providers who participated) may increase the theoretical transferability.

## Conclusion

While CMPs remain powerful tools for integrated care for people with complex needs, there is a persistent gap when it comes to fully engaging community stakeholders in case finding, as well as development and implementation of the individualized services plan. Formalized strategies to promote partnerships and better communication channels are needed, as well as the involvement of these stakeholders on governance committees at the healthcare system level.

## Supporting information

**S1 File. Interview guide.**
(DOC)

**S1 Checklist.**
(DOCX)

## Acknowledgments

We would like to thank the study participants, especially the community pharmacists and the CBO representatives. We also wish to thank Bonita Van Doorn for her revision of a previous version of this manuscript.

## Author Contributions

**Conceptualization:** Maud-Christine Chouinard.

**Data curation:** Mathieu Bisson.

**Formal analysis:** Maud-Christine Chouinard, Mathieu Bisson, Catherine Hudon.

**Funding acquisition:** Maud-Christine Chouinard.

**Methodology:** Maud-Christine Chouinard.

**Software:** Mathieu Bisson.

**Supervision:** Maud-Christine Chouinard, Catherine Hudon.

**Validation:** Maud-Christine Chouinard, Mathieu Bisson, Catherine Hudon.

**Writing – original draft:** Maud-Christine Chouinard, Mathieu Bisson, Catherine Hudon.

**Writing – review & editing:** Maud-Christine Chouinard, Mathieu Bisson, Alya Danish, Marlène Karam, Jérémie Beaudin, Nevena Grgurevic, Véronique Sabourin, Catherine Hudon.

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
