## [Decision Letter · Decision Letter 0]

5 Jul 2021

PONE-D-21-11296

Case management programs for people with complex needs: towards better engagement of community pharmacies and community-based organisations.

PLOS ONE

Dear Dr. Chouinard,

Thank you for submitting your manuscript to PLOS ONE. After careful consideration, we feel that it has merit but does not fully meet PLOS ONE’s publication criteria as it currently stands. Therefore, we invite you to submit a revised version of the manuscript that addresses the points raised during the review process.

We look forward to receiving your revised manuscript.

Kind regards,

Chaisiri Angkurawaranon

Academic Editor

PLOS ONE

Journal Requirements:

2. When reporting the results of qualitative research, we suggest consulting the COREQ guidelines or other relevant checklists listed by the Equator Network, such as the SRQR, to ensure complete reporting (http://journals.plos.org/plosone/s/submission-guidelines#loc-qualitative-research). In this case, please consider including more information on the number of interviewers, their training and characteristics; on how interviews and focus groups were conducted, and on participants selection and recruitment. Moreover, please provide the interview guide used as a Supplementary file

Reviewers' comments:

Reviewer's Responses to Questions

**Comments to the Author**

1. Is the manuscript technically sound, and do the data support the conclusions?

Reviewer #1: Yes

Reviewer #2: No

Reviewer #3: Partly

2. Has the statistical analysis been performed appropriately and rigorously? 

Reviewer #1: N/A

Reviewer #2: No

Reviewer #3: N/A

3. Have the authors made all data underlying the findings in their manuscript fully available?

Reviewer #1: No

Reviewer #2: No

Reviewer #3: No

4. Is the manuscript presented in an intelligible fashion and written in standard English?

Reviewer #1: Yes

Reviewer #2: Yes

Reviewer #3: No

5. Review Comments to the Author

Reviewer #1: Thank you for the opportunity to review this paper that reports on interviews and focus groups to explore the roles of community pharmacies and community-based organisations in the case management of patients with complex needs. This topic is relevant to an international audience considering the growing numbers of people with multiple co-morbidities and complex needs. The actual systems that are in place in various countries will differ, especially in terms of funding models, however the overall involvement of pharmacies and community support organisations would be relevant internationally.

General feedback

The manuscript is well written and flows well. My main comments relate to the need to incorporate more details in the methods section and the reason for the chosen methodology used for the analysis.

Introduction

The paragraph starting line 99 mentions the lack of knowledge about CBOs. There is literature showing lack of understanding of the role and scope of community pharmacies from consumers as well as health professionals that could be included here.

Minor:

Line 52: delete comma after episodic.

Line 91: this sentence does not read well. Should this be: … complex care needs such as those needing CMPs.?

Material and methods

The authors used descriptive qualitative design and include two references although both > 10 yrs (2000 and 2009). Why was this methodology followed? Reading the manuscript the process followed seemed to be similar to framework analysis. Could the authors add a statement on the chosen methodology?

Why do both focus groups and interviews? An explanation as to the reasoning behind using both would be useful.

Table 1 provides demographic details of patients however there is no details of the health professionals and managers in terms of years of practice experience and working in their various roles.

Minor: write out numbers <10

Results

More details are needed:

• When were the focus groups and interviews conducted?

• Who conducted the interviews and who facilitated the focus groups? Did the researcher have the appropriate skills?

• How long were interviews and focus groups (mean/median, min max)

• Were all interviews face-to-face?

• How was the interview tool developed and validated (face and content validation)?

Considering the number of interviews and focus groups conducted, I wanted to see more quotes to support the interpretation. I assume the interviews would have provided a wealth of data however only a few quotes are included to support the interpretation.

Discussion

The last sentence of the first paragraph: is there a gap between CBOs and community pharmacies or rather between these stakeholders and CMPs?

Reviewer #2: The aims of this studied were 1) to describe how case management programs engaged community pharmacies and community-based organisations in a perspective of integrated care for people with complex needs, and 2) to identify enablers, barriers and potential strategies for this engagement. However, there are no evidences derived from the scientific and statistical analyze in this study. The information of participants’ characteristics is lacking (e.g. comorbidity, reason for visiting to the emergency department or hospitalizations, drug history). What types of disease improved by using this program?

In addition, the authors should show the evidence of improving patients’ outcome. Did the number of hospitalization reduce by this program? How about the number of visiting to the emergency department? The study design is not sufficient to sustain the claims of the authors.

Reviewer #3: Summary of the research: This manuscript was a qualitative study examining how case management programs can better bridge/engage CBOs and community pharmacies to the health care system. The Taxonomy of Integrated Care was used to analyze the results. The results illustrate how different stakeholders view CMPs, CBOs, community pharmacies, etc., as well as the various components and stakeholders involved. Upon first read, I had a very difficult time following this paper and how the objective, conceptual model, methods, and results all tied together. Therefore, I would recommend substantial revisions to help enhance the clarity of this work.

Major issues:

1. In the introduction, with so many different players (CBOs, CMPs, community pharmacies, hospitals, clinics) that is was difficult to keep track of how they all tied together. It may help clarify this if there could be a figure of some sort. For example, CBOs and community pharmacies on one side, other stakeholder on the other, and CMPs forming a bridge between the two. Or CMPs serving as some type of umbrella. As it is now, it was not immediately apparent how all these pieces function in an ideal setting.

2. I was unfamiliar with the conceptual model. I think it could benefit from a sentence or two describing why that model was selected and how the model was developed. I think most readers will also be unfamiliar with the Rainbow Model of Integrated Care, so that requires a brief explanation. I also wonder if it would be more clearly described in a table where one column has the care processes and the second column has a description. This would make it easier to read and follow.

3. Why were both interview and focus groups carried out? This should be justified for the reader. Also, focus group size varied widely, what was the rationale for determining focus group size? Were focus group best practices followed? I think far more description of the methods is needed such as how participants were identified, how they were solicited to participate, their previous experience with CMPs, etc.

4. Analysis: I thought this work was guided by the Taxonomy for Integrated Care? Please clarify. I also think that more detail of the data analysis is needed. For example, can you describe in more detail what happened in each of the three stages? In addition, what is meant by cross-validation? Was there interrater reliability? Peer review? Member checking? I think more description is needed of how validity was maximized.

5. I found the results quite hard to follow. It may be better here to describe what each integration component means and how it appeared in the results of this study. Currently, the reader has to scroll between the conceptual model section and the results section to understand what each heading means. I also think it would benefit from more description (in other words, if all the quotes were removed, I think there should be adequate descriptions of each theme so that the reader understands how they presented in the results)

6. In the limitations, if participants had varying exposure to CMPs, it may have been helpful to organize the focus groups according to their exposure. For example, put all of those with high exposure in to the same groups and all of those with low exposure into the same groups. Since the authors state in the limitations that the Rainbow Model is difficult to use, there needs to be more justification for why it was selected.

7. It is difficult to understand how the results link to the objectives. It may be better to frame the results in terms of engagement strategies, engagement barriers, and engagement enablers to better answer the objective than group the results using the Rainbow Model. Alternatively, these engagement pieces can more clearly be described within each Rainbow Model component.

8. The objectives seem somewhat distinct. For example, at first why patients and physicians were being interviewed as they don't seem to be needed to address the first objective. Therefore, if the intent is to answer objective one, then it may be necessary to exclude data from patients, nurses, physicians.

Minor issues:

1. In the introduction, I would define CMPs similar to how you define integrated care

2. I would suggest consulting with a pharmacist on how to describe community pharmacy services as I am not sure what is meant by "file analysis"

3. There were a lot of acronyms making it hard to keep track of what they stood for. I would suggest writing more of them out.

4. What is a "descriptive qualitative design"? I would suggest describing this to the reader

5. p. 9 line 168 – what stakeholders were typically involved in the meeting?

6. I think including an interview guide would be helpful as the broad interview topics that are provided in the appendix to not give much insight on what was discussed.

6. PLOS authors have the option to publish the peer review history of their article (what does this mean?). If published, this will include your full peer review and any attached files.

Reviewer #1: No

Reviewer #2: No

Reviewer #3: No

---

## [Author Response · Author response to Decision Letter 0]

24 Sep 2021

JOURNAL REQUIREMENTS

Response: We standardized the manuscript (file naming and headings) according to PLOS ONE’s style requirements.

2. When reporting the results of qualitative research, we suggest consulting the COREQ guidelines or other relevant checklists listed by the Equator Network, such as the SRQR, to ensure complete reporting (http://journals.plos.org/plosone/s/submission-guidelines#loc-qualitative-research). In this case, please consider including more information on the number of interviewers, their training and characteristics; on how interviews and focus groups were conducted, and on participants selection and recruitment. Moreover, please provide the interview guide used as a Supplementary file

Response: Following the SRQR Checklist (https://www.equator-network.org/reporting-guidelines/srqr/), we added all the required information about the methods, especially the data collection (pages 10-11, lines 196-213, 220-235). We also included the Interview Guide and the completed SRQR Checklist as supplementary files.

a) If there are ethical or legal restrictions on sharing a de-identified data set, please explain them in detail (e.g., data contain potentially identifying sensitive patient information) and who has imposed them (e.g., an ethics committee). Please also provide contact information for a data access committee, ethics committee, or other institutional body to which data requests may be sent.

Response: We added information about ethical or legal restrictions on sharing our de-identified data set in the revised cover letter. 

Response: We included the captions at the end of the manuscript. 

REVIEW COMMENTS TO THE AUTHOR

Reviewer #1

Comment: Thank you for the opportunity to review this paper that reports on interviews and focus groups to explore the roles of community pharmacies and community-based organisations in the case management of patients with complex needs. This topic is relevant to an international audience considering the growing numbers of people with multiple co-morbidities and complex needs. The actual systems that are in place in various countries will differ, especially in terms of funding models, however the overall involvement of pharmacies and community support organisations would be relevant internationally.

Response: Thank you for this positive comment.

General feedback

Comment: The manuscript is well written and flows well. My main comments relate to the need to incorporate more details in the methods section and the reason for the chosen methodology used for the analysis.

Response: We added more information in the Methods section to further describe and explain the methodological choices (pages 9-13, lines 162-165, 185-186, 196-213, 220-235, 248-250, 252-263).

Introduction

Comment: The paragraph starting line 99 mentions the lack of knowledge about CBOs. There is literature showing lack of understanding of the role and scope of community pharmacies from consumers as well as health professionals that could be included here.

Response: We added a statement and a reference to this effect (page 6, line 109).

Minor

Comment: Line 52: delete comma after episodic.

Response: We made the change (page 4, line 54).

Comment: Line 91: this sentence does not read well. Should this be: … complex care needs such as those needing CMPs.?

Response: We moved "such as CMPs" earlier in the sentence to clarify its meaning (page 5; line 94).

Material and methods

Comment: The authors used descriptive qualitative design and include two references although both > 10 yrs (2000 and 2009). Why was this methodology followed? 

Response: We already explained the use of this approach (page 8; lines 161-163). However, we replaced the references with more recent ones, and we added an additional explanation for the rationale (page 8-9; lines 163-165). 

Comment: Reading the manuscript, the process followed seemed to be similar to framework analysis. Could the authors add a statement on the chosen methodology?

Response: The framework analysis method sits within a broad family of qualitative analysis methods, as does the thematic analysis. Although this method might have been relevant, it can be time-consuming and holds the potential for researchers to move away from the raw data (Ward DJ, Furber C, Tierney S, Swallow V. Using Framework Analysis in nursing research: a worked example. J Adv Nurs. 2013 Nov;69(11):2423-31). We added a statement about the chosen methodology in the Analysis section (page 13, lines 248-250).

Comment: Why do both focus groups and interviews? An explanation as to the reasoning behind using both would be useful.

Response: We added a brief explanation (pages 10-11, lines 208-213). 

Comment: Table 1 provides demographic details of patients however there is no details of the health professionals and managers in terms of years of practice experience and working in their various roles.

Response: Unfortunately, we did not collect these data. 

Comment: Minor: write out numbers <10

Response: We made the modifications (pages 9, 11, lines 175, 218).

Results

More details are needed:

Comment: When were the focus groups and interviews conducted?

Response: We added this information (page 10, line 197).

Comment: Who conducted the interviews and who facilitated the focus groups? Did the researcher have the appropriate skills?

Response: We specified this information (page 11, lines 227-229).

Comment: How long were interviews and focus groups (mean/median, min-max)

Response: The interviews lasted one hour. We added the duration of the focus groups, i.e. between 45 and 90 minutes (page 11, lines 216-217).

Comment: Were all interviews face-to-face?

Response: Yes, all interviews were face-to-face (page 11, line 227) 

Comment: How was the interview tool developed and validated (face and content validation)?

Response: We added this information (page 11; line 229-233). 

Comment: Considering the number of interviews and focus groups conducted, I wanted to see more quotes to support the interpretation. I assume the interviews would have provided a wealth of data however only a few quotes are included to support the interpretation.

Response: Indeed, a lot of data was collected with interviews and focus groups. We have voluntarily chosen to limit the number of quotations to leave more room for the text. However, we added more quotations to enrich the interpretation and to enhance the message (page 18, lines 300-308; page 21, lines 370-372; page 23, lines 429-433).

Discussion

Comment: The last sentence of the first paragraph: is there a gap between CBOs and community pharmacies or rather between these stakeholders and CMPs?

Response: We modified the sentence to explain that there is a gap between stakeholders and CMPs (page 24, line 445).

Reviewer #2

Comment: The aims of this studied were 1) to describe how case management programs engaged community pharmacies and community-based organisations in a perspective of integrated care for people with complex needs, and 2) to identify enablers, barriers and potential strategies for this engagement. However, there are no evidence derived from the scientific and statistical analysis in this study. 

Response: In qualitative research, there is no statistical analysis. The scientific literature on similar topics used similar methods and provided results comparable to ours. We used methods of qualitative data collection (individual interviews and focus groups) and qualitative data analysis (thematic analysis) largely recognized and validated in qualitative research.

Comment: The information of participants’ characteristics is lacking (e.g. comorbidity, reason for visiting the emergency department or hospitalizations, drug history). What types of diseases improved by using this program? In addition, the authors should show evidence of improving patients’ outcomes. Did the number of hospitalizations reduce by this program? How about the number of visits to the emergency department? The study design is not sufficient to sustain the claims of the authors.

Response: As already mentioned, the objectives of the study were 1) to describe how case management programs engaged community pharmacies and community-based organizations in a perspective of integrated care for people with complex needs, and 2) to identify enablers, barriers and potential strategies for this engagement. We did not aim to describe the characteristics of frequent users recruited in CMPs, neither to assess the impact of the CMPs on the patients’ health status or health service utilization. These outcomes are available in other studies mentioned in the introduction (page 4; lines 60 and 65).

Reviewer #3

Comment: Summary of the research: This manuscript was a qualitative study examining how case management programs can better bridge/engage CBOs and community pharmacies to the health care system. The Taxonomy of Integrated Care was used to analyze the results. The results illustrate how different stakeholders view CMPs, CBOs, community pharmacies, etc., as well as the various components and stakeholders involved. Upon first read, I had a very difficult time following this paper and how the objective, conceptual model, methods, and results all tied together. Therefore, I would recommend substantial revisions to help enhance the clarity of this work.

Response: Thank you for your comment. We made significant modifications to the manuscript to improve its readability and to clarify the concordance between the objectives, the conceptual model, the methods, and the results.

Major issues:

Comment: 1. In the introduction, with so many different players (CBOs, CMPs, community pharmacies, hospitals, clinics) that it was difficult to keep track of how they all tied together. It may help clarify this if there could be a figure of some sort. For example, CBOs and community pharmacies on one side, other stakeholders on the other, and CMPs forming a bridge between the two. Or CMPs serving as some type of umbrella. As it is now, it was not immediately apparent how all these pieces function in an ideal setting.

Response: Indeed, CMPs involve many actors with different roles and it is easy to get lost. We added Figure 1, entitled “Relationships between stakeholders engaged in CMPs”, as you suggested (page 6, line 104).

Comment: 2. I was unfamiliar with the conceptual model. I think it could benefit from a sentence or two describing why that model was selected and how the model was developed. I think most readers will also be unfamiliar with the Rainbow Model of Integrated Care, so that requires a brief explanation. I also wonder if it would be more clearly described in a table where one column has the care processes and the second column has a description. This would make it easier to read and follow.

Response: We agree that the conceptual framework may be difficult to understand in the current version and would need more explanation. To clarify, we added more details about the Rainbow Model of Integrated Care (RMIC) and its associated taxonomy, we described briefly the methods used for its development and we explained the choice of this model (page 7, lines 127-136). Rather than add a new table on the integration dimensions with each description, we prefer to refer the reader to the article by Valentijn et al. (2013), which includes a table with this information. However, we have included short definitions of the dimensions in the new Table 2, entitled “Strategies, enablers, and barriers for community stakeholders’ engagement in CMPs according to the Rainbow Model of Integrated Care” (pages 15-16), which we have added to the results section following your comments.

Comment: 3. Why were both interviews and focus groups carried out? This should be justified for the reader. 

Response: We added an explanation for the use of both individual interviews and focus groups (pages 10-11, lines 208-213).

Comment: Also, focus group size varied widely, what was the rationale for determining focus group size? Were focus group best practices followed? 

Response: We added a statement to justify the variation in the size of focus groups (page 11; lines 220-226). We also added information related to focus group best practices (pages 11-12; lines 227-235).

Comment: I think far more description of the methods is needed such as how participants were identified, how they were solicited to participate, their previous experience with CMPs, etc.

Response: We added more details on the data collection (page 10, lines 196-206). However, we cannot provide a description of the participants’ previous experience of the program because this information was not collected.

Comment: 4. Analysis: I thought this work was guided by the Taxonomy for Integrated Care? Please clarify. 

Response: As mentioned in the Conceptual framework section, both taxonomy and conceptual framework (Rainbow Model of Integrated Care) were used. We clarify it in the Analysis section (pages 13, lines 252-254).

Comment: I also think that more detail of the data analysis is needed. For example, can you describe in more detail what happened in each of the three stages? 

Response: We added a more detailed description of the three stages in the Analysis section (page 15, lines 252-259). 

Comment: In addition, what is meant by cross-validation? Was there interrater reliability? Peer review? Member checking? I think more description is needed of how validity was maximized.

Response: We replaced “cross-validation” with “investigator triangulation” (page 13; line 259), a notion that is more appropriate because it involves the participation of two or more researchers in the analysis processes to provide multiple observations and conclusions.

Comment: 5. I found the results quite hard to follow. It may be better here to describe what each integration component means and how it appeared in the results of this study. Currently, the reader has to scroll between the conceptual model section and the results section to understand what each heading means. I also think it would benefit from more description (in other words, if all the quotes were removed, I think there should be adequate descriptions of each theme so that the reader understands how they presented in the results)

Response: We developed a new table to present the results in a clearer and summarized form. Table 2, entitled “Strategies, barriers, and facilitators for community stakeholders’ engagement in CMPs according to the Rainbow Model of Integrated Care” (pages 15-16), aims to clarify the strategies, barriers, and enablers according to each integration dimension of the conceptual framework. We included short definitions of these dimensions in the left column.

Comment: 6. In the limitations, if participants had varying exposure to CMPs, it may have been helpful to organize the focus groups according to their exposure. For example, put all of those with high exposure into the same groups and all of those with low exposure into the same groups. 

Response: This is a good point. However, it was difficult to put stakeholders with the same level of exposure in the same groups because of their limited availability, especially the family physicians and the managers. Furthermore, it was not possible for the research team to know the level of the participants’ exposure to the CMPs before organizing the focus groups. Therefore, we preferred to gather the participants in a more “organic” way, i.e. according to their role and occupation.

Comment: Since the authors state in the limitations that the Rainbow Model is difficult to use, there needs to be more justification for why it was selected.

Response: We added a rationale about the choice of this model in the Conceptual framework section and in the Limitations (page 7, lines 127-136; page 26, lines 489-492). 

Comment: 7. It is difficult to understand how the results link to the objectives. It may be better to frame the results in terms of engagement strategies, engagement barriers, and engagement enablers to better answer the objective than group the results using the Rainbow Model. Alternatively, these engagement pieces can more clearly be described within each Rainbow Model component.

Response: As mentioned, we developed Table 2 (page 15-16) to organize more clearly the identified strategies, barriers, and enablers for community stakeholders’ engagement in CMPs. Short definitions can be found under each integration dimension of the conceptual framework.

Comment: 8. The objectives seem somewhat distinct. For example, at first why patients and physicians were being interviewed as they don't seem to be needed to address the first objective. Therefore, if the intent is to answer objective one, then it may be necessary to exclude data from patients, nurses, physicians.

Response: Although the main project interview guide did not include a question specific to Objective 1 of this subproject for patients, nurses and physicians, their transcripts might have contained testimonials about their connections to community agencies and pharmacies related to the CMPs. We then decided to review the transcripts from these participants to ensure that we get the most complete description possible. 

Minor issues:

Comment: 1. In the introduction, I would define CMPs similar to how you define integrated care

Response: Even if these two notions can seem similar, case management refers to ‘a collaborative, client-driven process for the provision of quality health and support services through the effective and efficient use of resources’. We added this definition (page 4; lines 60-62) to distinguish CMPs from the definition of ‘integrated care’.

Comment: 2. I would suggest consulting with a pharmacist on how to describe community pharmacy services as I am not sure what is meant by "file analysis"

Response: We changed “file analysis” to “medical information review” to clarify (page 5, line 85).

Comment: 3. There were a lot of acronyms making it hard to keep track of what they stood for. I would suggest writing more of them out.

Response: We have limited the number of acronyms by replacing IUHSSC with 'hospital network' (pages 9, 10, 11, lines 170, 177, 180, 203), HSSC with 'hospital' (pages 9, 10, 18, 19, 21, 22, 23, 25, lines 180, 197, 296, 313, 316, 334, 368, 378, 394, 411, 415, 472), CMPs with 'programs' in some places (pages 6, 7, 9, 21, 25, 26 lines 102, 110, 117, 180, 377, 452, 458, 478, 486) and we spelled out ISP throughout (pages 9, 10, 17, 19, 20, 21, 22, 25, 26, lines 183, 189, 269, 273, 277, 281, 314, 317, 322, 326, 335, 349, 362, 391, 455, 459, 477).

Comment: 4. What is a "descriptive qualitative design"? I would suggest describing this to the reader

Response: We added a statement to further describe this method (page 9; lines 163-165). 

Comment: 5. p. 9 line 168 – what stakeholders were typically involved in the meeting?

Response: We added this information (page 9; lines 185-186).

Comment: 6. I think including an interview guide would be helpful as the broad interview topics that are provided in the appendix to not give much insight on what was discussed.

Response: We added the Interview Guide as a supplementary file, grouping the questions by participant category.

---

## [Decision Letter · Decision Letter 1]

22 Nov 2021

Case management programs for people with complex needs: towards better engagement of community pharmacies and community-based organisations.

PONE-D-21-11296R1

Dear Dr. Chouinard,

We’re pleased to inform you that your manuscript has been judged scientifically suitable for publication and will be formally accepted for publication once it meets all outstanding technical requirements.

Kind regards,

Chaisiri Angkurawaranon

Academic Editor

PLOS ONE

Additional Editor Comments (optional):

Reviewers' comments:

Reviewer's Responses to Questions

**Comments to the Author**

1. If the authors have adequately addressed your comments raised in a previous round of review and you feel that this manuscript is now acceptable for publication, you may indicate that here to bypass the “Comments to the Author” section, enter your conflict of interest statement in the “Confidential to Editor” section, and submit your "Accept" recommendation.

Reviewer #3: All comments have been addressed

2. Is the manuscript technically sound, and do the data support the conclusions?

Reviewer #3: Yes

3. Has the statistical analysis been performed appropriately and rigorously? 

Reviewer #3: N/A

4. Have the authors made all data underlying the findings in their manuscript fully available?

Reviewer #3: No

5. Is the manuscript presented in an intelligible fashion and written in standard English?

Reviewer #3: Yes

6. Review Comments to the Author

Reviewer #3: Thank you for your careful review of feedback and addressing all of the provided comments. No further edits are needed in my opinion

7. PLOS authors have the option to publish the peer review history of their article (what does this mean?). If published, this will include your full peer review and any attached files.

Reviewer #3: No

---

## [Editor Report · Acceptance letter]

29 Nov 2021

PONE-D-21-11296R1 

Case management programs for people with complex needs: towards better engagement of community pharmacies and community-based organisations. 

Dear Dr. Chouinard:

I'm pleased to inform you that your manuscript has been deemed suitable for publication in PLOS ONE. Congratulations! Your manuscript is now with our production department. 

Kind regards, 

on behalf of

Dr. Chaisiri Angkurawaranon 

Academic Editor

PLOS ONE